# Systematic Evaluation of LLM-as-a-Judge in LLM Alignment Tasks: Explainable Metrics and Diverse Prompt Templates

**Hui Wei**[*†1], **Shenghua He**[*2], **Tian Xia**[2], **Fei Liu**[3], **Andy Wong**[†4], **Jingyang Lin**[†5], **Mei Han**[2]
[1]UC Merced, [2]PAII Inc., [3]Emory University, [4]Inflection AI, [5]University of Rochester

 Code: `https://github.com/shenghh2015/llm-judge-eval`

## Abstract

LLM-as-a-Judge has been widely applied to evaluate and compare different LLM alignmnet approaches (e.g., RLHF and DPO). However, concerns regarding its reliability have emerged, due to LLM judges' biases and inconsistent decision-making. Previous research has developed evaluation frameworks to assess reliability of LLM judges and their alignment with human preferences. However, the employed evaluation metrics often lack adequate explainability and fail to address LLM internal inconsistency. Additionally, existing studies inadequately explore the impact of various prompt templates when applying LLM-as-a-Judge methods, leading to potentially inconsistent comparisons between different alignment algorithms. In this work, we systematically evaluate LLM-as-a-Judge on *alignment tasks* by defining more theoretically interpretable evaluation metrics and explicitly mitigating LLM internal inconsistency from reliability metrics. We develop an open-source framework to evaluate, compare, and visualize the reliability and alignment of LLM judges, which facilitates practitioners to choose LLM judges for alignment tasks. In the experiments, we examine effects of diverse prompt templates on LLM-judge reliability and also demonstrate our developed framework by comparing various LLM judges on two common alignment datasets (i.e., TL;DR Summarization and HH-RLHF-Helpfulness). Our results indicate *a significant impact of prompt templates on LLM judge performance*, as well as *a mediocre alignment level between the tested LLM judges and human evaluators*.

## 1 Introduction

Commercial LLMs (e.g., GPT-4 (Achiam et al., 2023)) have been widely used as the surrogates for human evaluators, referred to as LLM-as-a-Judge (Gu et al., 2024; Li et al., 2024a;b), to perform pairwise evaluation on numerous LLM alignment tasks, such as summarization and multiturn conversations. Since these commercial models have already been extensively trained with advanced alignment techniques (Achiam et al., 2023; Touvron et al., 2023), they are capable of approximating human preferences (Rafailov et al., 2024b; Zheng et al., 2024).

While it is plausible to utilize these models as surrogates for human judges, biases and inconsistencies are frequently observed in their judgment results, despite the application of various bias-mitigation techniques (Rafailov et al., 2024a;b). This necessitates a systematic investigation of LLM judge reliability and alignment with human preferences in the context of LLM alignment tasks.

Previous studies have evaluated LLM-as-a-Judge methods on various language generation tasks (Wang et al., 2023b; Saito et al., 2023; Dubois et al., 2024a; Panickssery et al., 2024; Shi et al., 2024; Thakur et al., 2024; Chiang & Lee, 2023; Liu et al., 2023; Wang et al., 2023a; Zhu et al.; Wang et al., 2023c; Li et al., 2023; Zheng et al., 2024; Li et al., 2024c; Chiang et al., 2024; Dubois et al., 2024b). However, these studies encounter three main limitations:

---

[*]Equal contributions & Corresponding authors:  huiwei2@ucmerced.edu,  shenghh2015@gmail.com
[†]Work was done when H. Wei, J. Lin, and A. Wong were at PAII Inc.

- Lacking theoretical interpretability for bias definitions (e.g. position bias and length bias).
- Not considering internal inconsistencies (i.e., system noise) by assuming LLM judges make deterministic decisions across identical experiments.
- Focusing on evaluating various LLMs, while the effects of prompt templates have been insufficiently examined.

In this study, we aim to address these limitations and advance the systematic evaluation of LLM-as-a-Judge on LLM alignment tasks. Our main contributions in this work are:

- We improve the theoretical explainability of evaluation metrics for assessing LLM-judge position and length bias, by (1) defining them within a unified accuracy-based framework, (2) explicitly defining the LLM internal self-inconsistency as *flipping noise* and mitigating its impact, as well as (3) formally analyzing the relationship between position and length bias after mitigation.
- We develop a open-sourced framework to evaluate, compare, and visualize the alignment and reliability of LLM judges, allowing for a wide range of LLMs and user-defined prompt templates. In the experiments, we leverage the developed framework to test *a wide range of prompt templates with diverse formats* and investigate their impact on LLM judge performance.
- Our results indicate *a significant impact of prompt templates on LLM judge performance*, underscoring the need for a thorough and careful comparison of various LLMs and prompt templates before employing the LLM-as-a-Judge methodology.

## 2 BACKGROUND AND RELATED WORK

In this section, we define the pairwise evaluation task conducted by both human and LLM judges, and examine self-inconsistency and biases inherent in LLM judges. Additionally, we review relevant literature on position bias and length bias.

**Human-based Pairwise Evaluation** Given a set of $N$ questions, each paired with responses generated by separate LLMs, the human judge is asked to select the better response based on predefined criteria, such as coherence and helpfulness. Let $N_1$ and $N_2$ be the numbers that the first and second answer are chosen. The win rate of the first and the second LLM is defined as $w_{1,2} = \frac{N_1}{N}$ and $w_{2,1} = 1 - w_{1,2} = \frac{N_2}{N}$.

**LLM-based Pairwise Evaluation** LLM judges are subjected to the same evaluation procedures as human judges. However, compared with humans, LLMs are more sensitive to instructions (i.e., prompt templates) (Stureborg et al., 2024; Zhu et al.). Thus, in this study, we define an LLM-judge as *the combination of a specific LLM and a particular prompt template*.

**LLM-Judge Self-Inconsistency** Previous studies have observed that LLM judges (Shi et al., 2024; Stureborg et al., 2024) may produce inconsistent judgments even when presented with identical prompts. This is caused by non-greedy decoding strategies leveraged by LLMs, such as *top-p* and *top-k*, which generate non-deterministic outputs. The non-deterministic level is controlled by the *temperature* parameter. In this work, we refer to these inconsistencies as self-inconsistency or system noise in LLM judges and model and quantify them using the term *flipping noise* (Section 4).

**LLM-Judge Position and Length Bias** *Positioin bias* refers to LLM-judge's systematic preference for a specific response position (the first or the second in the pairwise evaluation task). Wang et al. (2023b) and Lee et al. (2023) observed the position bias when using GPT-4 (Achiam et al., 2023) and PaLM 2 (Anil et al., 2023) as the judge for the pairwise comparison between candidate LLMs. They measured the position bias by the ratio of inconsistent decisions made by LLM judges after swapping response positions. Differently, Liusie et al. (2023) and Zheng et al. (2024) defined the position bias as the disparity of selection probabilities after reversing the response order. *Length bias* refers to LLM judge's systematic preference for longer responses even when their qualities are similar to shorter versions. Saito et al. (2023) observed a discrepancy between LLMs and human preferences regarding response length. They employed accuracy parity—related to human preferences for longer responses and shorter responses—to measure relative length bias.

In contrast to above studies, our work theorectically examines the impact of LLM judge self-inconsistency on position bias and length bias metrics, and provides practical methods to mitigate this effect. We also provide a theoretical analysis and validation of our defined metrics to enhance their interpretability. Additionally, we investigate the relationship between these biases and accuracy, revealing significant insights. Finally, our study includes an extensive evaluation of position and length bias across a diverse set of LLM judges with various prompt templates.

## 3 NOTATION

Let $\mathcal{D} = \{h_n | n = 1 \ldots N\}$ be a human-preference dataset containing $N$ data cases. An individual data case $h_n = (x^{(n)}, y_c^{(n)}, y_r^{(n)})$ represents a prompt-responses pair with a human preference label, where $x^{(n)}$ is a prompt (e.g., a post for summarization), $y_c^{(n)}$ is the preferred LLM response and $y_r^{(n)}$ is the less preferred response, both by human evaluators. We assume each case is drawn from the distribution $h_n \sim p(h|\theta)$, where $\theta$ represents the underlying human preferences depending on human annotators helping construct the dataset. We drop the data case index $n$ for brevity when the context is clear.

## 4 EXPLAINABLE EVALUATION METRICS AND FLIPPING NOISE

**Accuracy** Accuracy measures the alignment level of LLM judges with human preferences. Formally, we denote $\theta_l$ as the underlying preference by some LLM-judge $l$, and accuracy evaluates how closely $\theta_l$ is to $\theta$, where $\theta$ is the human preference defined in Section 3.

There are two versions of the accuracy metric: $\text{Acc}_{\text{both}}$ and $\text{Acc}_{\text{random}}$. We assume the LLM judge decides on each data case by considering two response orders: $h = (x, y_c, y_r)$ and $h' = (x, y_r, y_c)$. The LLM judge then selects the preferred response $y$ and $y'$ from each order $h$ and $h'$, where $y, y' \in \{y_c, y_r\}$. Broadly, we denote the set of LLM judging results as $J = \{s_n | n = 1 \ldots N\}$, where each result $s_n = (y^{(n)}, y'^{(n)})$ represents the selection outcome from both response orders across all the data cases in the dataset $\mathcal{D}$. Then the accuracy metrics $\text{Acc}_{\text{both}}$ and $\text{Acc}_{\text{random}}$ can be defined over the judging set $J$ as follows:

$$\text{Acc}_{\text{both}} = \frac{1}{N} \sum_{n=1}^{N} \mathbb{1}\left(y^{(n)} = y_c^{(n)} \wedge y'^{(n)} = y_c^{(n)}\right), \qquad \text{Acc}_{\text{random}} = \frac{1}{N} \sum_{n=1}^{N} \mathbb{1}\left(y_{\text{random}}^{(n)} = y_c^{(n)}\right)$$

where $y_{\text{random}}$ is randomly chosen from $\{y, y'\}$ with the probability of 0.5.

**Flipping Noise** As mentioned in Section 2, LLM outputs are generally non-deterministic, which can lead to inconsistent judgments even when the *same* LLM judge is presented with the *identical* data case $h = (x, y_c, y_r)$. To better model this behavior, we first assume the LLM judge's outputs are always *deterministic* (i.e., no self-inconsistency), representing its decision as a binary variable $X \in \{0, 1\}$, where $X = 1$ indicates LLM judge's selection of the human-preferred response $y_c$ and $X = 0$ indicates otherwise. Under this assumption, re-evaluating the same case $h$ would always yield the same decision (e.g., selecting $y_c$ with $X = 1$).

However, if we *consider* self-inconsistency, the LLM judge may instead select the alternative response $y_r$ upon re-evaluation. We refer to this as "flipping" its original decision. **We define the discrepancy between the LLM judge's actual decision (considering self-inconsistency) and its original decision (assuming no self-inconsistency) as *flipping noise*, which quantifies the impact of self-inconsistency.** We introduce another binary variable, $Z \in \{0, 1\}$, to represent the LLM judge's actual decision, which may differ from its original value $X$ due to flipping noise. In real-world scenarios where self-inconsistency is *unavoidable*, *we can only observe the noisy decision $Z$, not the idealized $X$.*

Formally, we can represent the relationship between LLM judge's original decision and actual decision as follows:

$$Z = \begin{cases} 1 - X, & p\left[1 - X | X\right] = q \\ X, & p\left[X | X\right] = 1 - q \end{cases} \tag{1}$$

where $q$ is the probability that the LLM judge's decision is flipped. For a completely deterministic LLM judge, $q = 0$.

**Position Bias (PB)** As a reminder, we define accuracy based on two sets of responses with reversed orders, namely $(y_c, y_r)$ and $(y_r, y_c)$, for the same prompt $x$. To assess accuracy, we require the LLM judge to be evaluated in both orders. Here, we employ the same setting to define position bias.

First of all, we define $p[X = 1|(y_c, y_r)]$ as the probability that the LLM-judge's *original* result aligns with the human selection for the response order $(y_c, y_r)$, and $p[X = 1|(y_r, y_c)]$ as the probability that the LLM-judge's result aligns with the human selection when the order is reversed. It is important to note these two probabilities are essentially *accuracy* metrics for the two response positions.

We first consider a special case where the LLM judge makes a *fully consistent decision* (i.e. $q = 0$, $Z = X$), and is *completely insensitive to the response position order* (i.e. exhibits no position bias). This implies that accuracy should be invariant regarding response positions: $p[X = 1|(y_c, y_r)] - p[X = 1|(y_r, y_c)] = 0$.

Additionally, if the LLM-judge *exhibits position bias favoring the first position over the second*, it will select $y_c$ more frequently in $(y_c, y_r)$ and $y_r$ more frequently in $(y_r, y_c)$, compared to the scenario with no position bias. Thus, the accuracy $p[X = 1|(y_c, y_r)]$ will increase and $p[X = 1|(y_r, y_c)]$ will decrease, resulting in $p[X = 1|(y_c, y_r)] - p[X = 1|(y_r, y_c)] > 0$. The same rationale applies when the second position is preferred.

Based on these intuitions, **we define position bias as**:

$$\text{PB} = p[X = 1|(y_c, y_r)] - p[X = 1|(y_r, y_c)] \tag{2}$$

where the absolute value $|\text{PB}|$ measures the degree of position bias, with positive and negative values indicating preferences for the first and second positions, respectively.

Finally, we address the general case in which the LLM-judge *makes non-deterministic decisions and exhibits position bias*. Here, only noisy observation $Z$ defined in Eq. 1 is observable, instead of $X$. Thus, to mitigate the impact of self-inconsistency and determine the original underlying position bias as defined by Eq. 2, we first compute the accuracy of both positions based on $Z$, and then apply the de-noise process according to the following relationships between accuracy based on $X$ and accuracy based on $Z$.

$$p[X = 1|(y_c, y_r)] = \frac{p[Z = 1|(y_c, y_r)] - q_{cr}}{1 - 2 \cdot q_{cr}}, \ q_{cr} = p[1 - X|X, (y_c, y_r)]$$

$$p[X = 1|(y_r, y_c)] = \frac{p[Z = 1|(y_r, y_c)] - q_{rc}}{1 - 2 \cdot q_{rc}}, \ q_{rc} = p[1 - X|X, (y_r, y_c)]$$

where $q_{cr}$ and $q_{rc}$ are the probabilities that the LLM judge's decision is flipped for response order $(y_c, y_r)$ and $(y_r, y_c)$, respectively. In the appendix A.4, we derive the above relationships, validate the position bias measurement based on de-noised accuracies, and provide a practical method for their computation.

**Length Bias (LB)** Previous studies have indicated that human evaluators exhibit the length bias when assessing responses (Zheng et al., 2024; Saito et al., 2023). If LLM judges are employed as surrogates for human judges, it is expected they have the same length bias in general. Thus, this study aims to measure the *relative* length bias of LLM-judges compared with human evaluators, rather than their absolute length bias. For brevity, we use "length bias" to refer to the *relative* length bias in the paper.

For each data case $(x, y_c, y_r)$, we denote $\Delta l = l_c - l_r$ as the length difference between $y_c$ and $y_r$, where $l_c$ and $l_r$ are the length of $y_c$ and $y_r$, respectively. Additionally, we denote $p[X = 1|\Delta l > 0]$ as the probability that the LLM-judge's result aligns with the human selection when the human selected response $y_c$ is longer than $y_r$, and $p[X = 1|\Delta l \leq 0]$ as the probability that the LLM-judge's result align when the length relationship is reversed. Moreover, these two probabilities are defined within the same accuracy framework, analogous to the definition of position bias.

Following the same rationale as in the position bias section, **we define length bias as**

$$\text{LB} = p[X = 1|\Delta l > 0] - p[X = 1|\Delta l \leq 0] \tag{3}$$

where |LB| measures how significantly the LLM judge exhibits different length bias compared to human judges and the sign of LB indicates it biases more towards longer response or shorter responses than human judges, respectively.

In cases where *flipping noise cannot be neglected*, analogous to the approach for position bias, we first compute accuracies from noisy observations $Z$: $p\left[Z=1|\Delta l>0\right]$ and $p\left[Z=1|\Delta l\leq 0\right]$. We then apply a de-noising process to mitigate the impact of self-inconsistency based on the relationships between accuracy derived from $X$ and accuracy derived from $Z$ as follows:

$$p\left[X=1|\Delta l>0\right]=\frac{p\left[Z=1|\Delta l>0\right]-q_{\Delta l>0}}{1-2\cdot q_{\Delta l>0}},\ q_{\Delta l>0}=p\left[1-X|X,\Delta l>0\right]$$

$$p\left[X=1|\Delta l\leq 0\right]=\frac{p\left[Z=1|\Delta l\leq 0\right]-q_{\Delta l\leq 0}}{1-2\cdot q_{\Delta l\leq 0}},\ q_{\Delta l\leq 0}=p\left[1-X|X,\Delta l\leq 0\right]$$

where $q_{\Delta l>0}$ and $q_{\Delta l\leq 0}$ are the probabilities that the LLM judge's decision is flipped for the conditions $\Delta l>0$ and $\Delta l\leq 0$, respectively. In the appendix A.4, we derive the above relationships, validate the length bias measurement based on de-noised accuracies, and provide a practical method for their computation.

**Further Analysis** To enhance the interpretability of the position and length bias metrics, we further analyze their inter-relationship theoretically. Our findings are summarized below and formally proven in the appendix A.4.

**Finding 1** Position bias definition in Eq. 2 is intrinsically length bias-mitigated.

**Finding 2** Length bias measurement in Eq. 3 is entangled with position bias. Employing $A_{\text{both}}$ for accuracy helps mitigate the influence of positional bias in the assessment of length bias.

## 5 EVALUATION FRAMEWORK

In this study, we introduce an evaluation framework that integrates our proposed methods for computing metrics, including accuracy ($\text{Acc}_{\text{both}}$, $\text{Acc}_{\text{random}}$), position bias and length bias. **The framework is developed and open-sourced to help researchers and practitioners select either predefined or user-customized LLM judges for alignment tasks based on aforementioned evaluation metrics and their specific needs.**

The pipeline of the framework, as depicted in Fig. 1, is structured into four modular components: 1) *Data Sampler*, 2) *LLM Judges*, 3) *Metrics Computation*, and 4) *Metrics Visualization*. The functionality of each component is detailed as follows.

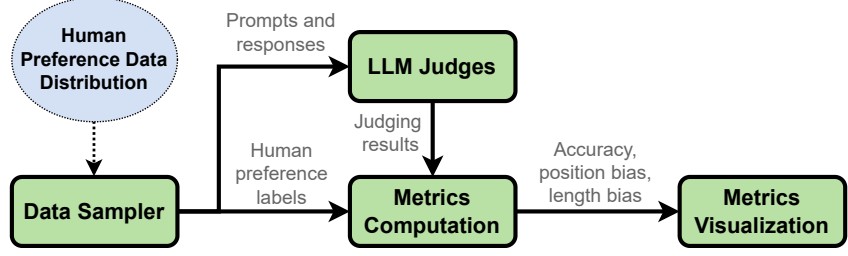

Figure 1: LLM-as-a-Judge Evaluation Framework

**Data Sampler:** When dealing with a large human preference dataset and a limited budget for using commercial LLM models, it becomes necessary to sample a manageable-size subset from the full dataset for LLM judge evaluation. Our framework employs a *stratified sampling strategy* to ensure that the subset maintains the same proportion of different conditions (e.g. length difference distribution) as the original dataset.

**LLM Judges:** As defined in Section 2, an LLM judge refers to the combination of a particular LLM and a specific prompt template. Given an LLM judge, this module is responsible for generating textual judging decisions for each sampled data case and subsequently converting them into a binary

outcome for metrics computation. This module allows the flexible creation of varied LLM judges by configuring different LLMs and prompt templates for evaluation.

**Metrics Computation:** This module computes alignment and reliability evaluation metrics (i.e. accuracy, position bias, and length bias) using the judging results from the LLM Judge module and the human preference labels provided by the dataset, based on the computational methods described in the Method section.

**Metrics Visualization:** This module visualizes both the *individual* computed metrics and their *inter-relationships*, providing comprehensive insights for comparing LLM judges and aiding in the selection of the most suitable LLM judge for specific LLM-alignment tasks.

## 6 EXPERIMENTS

**Data Selection** We demonstrate our evaluation framework using two datasets that are commonly used to evaluate LLM alignment algorithms: TL;DR summarization dataset (Völske et al., 2017; Stiennon et al., 2020) and HH-RLHF-Helpfulness dataset (Bai et al., 2022). Both datasets contain a prompt (a post for the summarization dataset; a conversation history between humans and LLM assistants for HH-RLHF dataset) with two responses generated by distinct LLMs for each sample. Also, human preference labels are available to indicate which response is more aligned with human preference. Both datasets have already been partitioned into train and test sets by the authors in the original studies.

In our experiments, it is highly time-consuming and expensive to evaluate LLM judges on all the data cases of both datasets (143,356 for summarization and 124,243 for HH-RLHF-Helpfulness), so we randomly sample a subset from each dataset to perform all the evaluation experiments. Compared with the summarization dataset, the HH-RLHF-Helpfulness dataset has a much smaller test set (6,240 vs. 70,228), thus, we select a subset from the TL;DR summarization *test* set following the previous study (Rafailov et al., 2024b) and a subset from the *entire* HH-RLHF-Helpfulness dataset. Moreover, multiple data cases may share the same prompt (post or conversation history) with distinct response pairs. To make our collected datasets as diverse as possible, only one pair is kept for this prompt and others are removed. After this step, each unique prompt corresponds to only one unique answer pair. Then we randomly sample the prompts and their associated responses five times without replacement, resulting in five non-overlapping splits. Since measuring length bias requires dividing all the data cases into two conditions: whether longer responses are preferred by humans or not, we leverage the stratified sampling to preserve the same ratio of these two conditions as in the entire dataset.

Overall, both datasets used in our experiments contain 200 distinct samples for each split, which results in 1000 samples in total. The summarization and HH-RLHF-Helpfulness datasets have a stratified ratio (# of humans prefer longer responses: # of humans prefer shorter responses) of 115:85 and 111:89 respectively.

**LLM Judges** Our LLM judges integrate a range of varied commercial large language models and prompt templates. Particularly, we assess **GPT-4o**, **GPT-4o-mini** and **GPT-3.5-turbo** with **8 templates** on the summarization dataset and **10 templates** on the HH-RLHF-Helpfulness dataset. Thus, there are $3 \times 8 = 24$ LLM judges for the summarization dataset and $3 \times 10 = 30$ LLM judges for the HH-RLHF-Helpfulness dataset.

GPT-4o is one of the most advanced models which has the latest checkpoint on 08/06/2024, GPT-4o-mini is the most cost-efficient model, while GPT-3.5-turbo is from the last OpenAI model generation and serves as the baseline in our experiments. Our preliminary studies suggest that GPT-4o exhibits comparable performance to GPT-4 in judging decision-making, but at a cost that is 4 to 6 times lower. Due to limited budget, we select GPT-4o for evaluation over GPT-4 from the list of *commercial* LLMs, despite GPT-4 being the most widely-used model in LLM alignment studies before the release of GPT-4o.

All the considered templates were actually used in the pairwise comparison tasks to evaluate different LLM alignment algorithms by papers of year 2023 and 2024, and we make sure they all have *dissimilar* prompt formats. Furthermore, since our evaluation datasets have no "tied" labels from human annotations, which indicate two responses are equally preferred, we remove sentences from

the prompt templates which allow LLM judges to select "tied" labels. Please refer to the appendix A.1 for template examples of each dataset, as well as a complete list of the papers from which all the templates in this study are derived.

**Temperature Parameter Selection** Temperature parameter determines how deterministic LLM outputs are, which might affect the performance of LLM-judges. However, few previous studies that use LLMs as judges explicitly explain how and why they choose the temperature in their experiments. In this study, we assess the impact of the temperature parameter on the self-consistency (i.e. 1-flipping probability $q$) and accuracies of the large language models, which helps to select the temperature before evaluating LLM-judge performance using other metrics.

In detail, we investigate five temperature settings: 0.0, 0.1, 0.3, 0.5, and 0.7. For each temperature setting, we concatenate data samples in all 5 splits (1000 samples in total) and repeatedly ask LLM judges to select the better response $K = 5$ times for each sample. We compute the self-consistency for both response positions $(y_c, y_r)$ and $(y_r, y_c)$ separately, as well as $\text{Acc}_{\text{both}}$ across all the samples.

Through preliminary experiments, we found the impact of different temperatures is the same to the *same* LLM with *different* prompt templates, so in the large-scale experiments, only the prompt templates from DPO paper (Rafailov et al., 2024b) are utilized for both datasets.

**Metrics Computation** To compute the **flipping probability**, same as selecting the temperature parameter, we let LLM judges select their preferred response from each sample repeatedly for $K = 5$ times. However, since we need to compute this probability for every LLM judge (24 for the summarization dataset and 30 for the HH-RLHF-Helpfulness dataset), we only leverage *the first split* of each dataset due to limited budget and assume they remain consistent on all five splits. For each sample, the flipping probabilities $q_{cr}$ and $q_{rc}$ for both positions $(y_c, y_r)$ and $(y_r, y_c)$ are computed separately to estimate de-noised position bias, and the flipping probabilities $q_{\Delta l > 0}$ and $q_{\Delta l \leq 0}$ are computed as well to calculate de-noised length bias. To compute **accuracy, position bias, and length bias**, we compute each metric on *all the splits* ($S = 5$). In the result, we report the mean and standard deviation of LLM judge performances across these five splits.

## 7 RESULTS

**Temperature** Table 1 contains the results of self-consistent rate (SCR) and accuracy with various temperatures. The self-consistent rate, given by $1 - q$ as defined in Eq. 1, measures the probability that the LLM's judgments are consistent across identical inputs. Since different LLMs show the same trend on both datasets, we only include GPT-4o here for the demonstration. Results regarding other LLMs are included in the appendix A.3.

From the table, we observe that *higher temperatures result in lower self-consistency for both positions, while accuracy is not significantly affected by temperatures*. Specifically, even when the temperature is set to 0.0, complete self-consistency (i.e. SCR=1.0) remains unachievable. Furthermore, *self-consistency varies with different positions*, thereby necessitating the separate measurement of flipping probabilities related to flipping noise associated with each position.

Finally, we aim to demonstrate the generalizability of our evaluation framework by employing a value that is not a special case, such as 0.0. Thus, *we select 0.1 as the temperature in all of our experiments*, which has the highest level of self-consistency compared with higher temperatures.

| | TL;DR Summarization | | | HH-RLHF-Helpfulness | | |
|---|---|---|---|---|---|---|
| **Temperature** | **SCR** $(y_c, y_r)$ | **SCR** $(y_r, y_c)$ | **Acc** $(\text{Acc}_{\text{both}})$ | **SCR** $(y_c, y_r)$ | **SCR** $(y_r, y_c)$ | **Acc** $(\text{Acc}_{\text{both}})$ |
| 0.0 | 0.977 | 0.971 | 0.665 (0.003) | 0.974 | 0.967 | 0.573 (0.005) |
| 0.1 | 0.973 | 0.967 | 0.666 (0.004) | 0.966 | 0.957 | 0.575 (0.005) |
| 0.3 | 0.963 | 0.956 | 0.668 (0.003) | 0.950 | 0.944 | 0.574 (0.005) |
| 0.5 | 0.953 | 0.949 | 0.663 (0.003) | 0.942 | 0.926 | 0.579 (0.009) |
| 0.7 | 0.946 | 0.927 | 0.657 (0.000) | 0.934 | 0.914 | 0.577 (0.006) |

Table 1: Self-consistent rate (SCR) and accuracy (Acc) of tested temperatures for the TL;DR summarization and HH-RLHF-Helpfulness datasets. Results are demonstrated using GPT-4o and prompt templates from the DPO paper Rafailov et al. (2024b).

**Accuracy** Figure 2a and Figure 2b show accuracies (Acc$_{both}$) of LLM judges on both datasets, where identical colors represent the same prompt template within the same dataset (the same coloring rule is applied to all the result figures except for Figure 4). As we can see, different LLM judges have distinct accuracy, which means they have varied alignment levels with human preferences. Also, it demonstrates *the performance of an LLM judge is highly sensitive to prompt templates.*

Notably, several LLM judges have very low accuracies (Acc$_{both}$ < 0.2). Thus, it is significantly important to carefully evaluate and compare different LLM judges before actually using them to evaluate LLM alignment algorithms. Moreover, we find that *all the accuracies on both datasets are below 0.7, which shows the mediocre alignment level and demonstrates that human evaluation is necessary to precisely compare different LLM alignment systems.*

Compared with GPT-3.5-turbo, both GPT-4o and GPT-4o-mini have higher accuracies no matter which prompt template is used. It demonstrates that the superior internal capacities of recent LLMs, compared to older versions, are independent of the prompt templates used.

Figure 2c and Figure 2d show accuracy (Acc$_{random}$) of LLM judges on both datasets. Compared with Figure 2a and 2b (i.e. Acc$_{both}$), the gap between GPT-3.5-turbo and the others shrinks. This is because Acc$_{random}$ involves randomly selecting a position when LLM judge selection is inconsistent across two positions, thereby not reflecting the internal capabilities of LLM judges. Consequently, *Acc$_{random}$ is a less effective metric for assessing LLM judge performance compared to Acc$_{both}$.* Based on this, only Acc$_{both}$ is used to demonstrate the relationship between accuracy and position bias as well as length bias in the following sections.

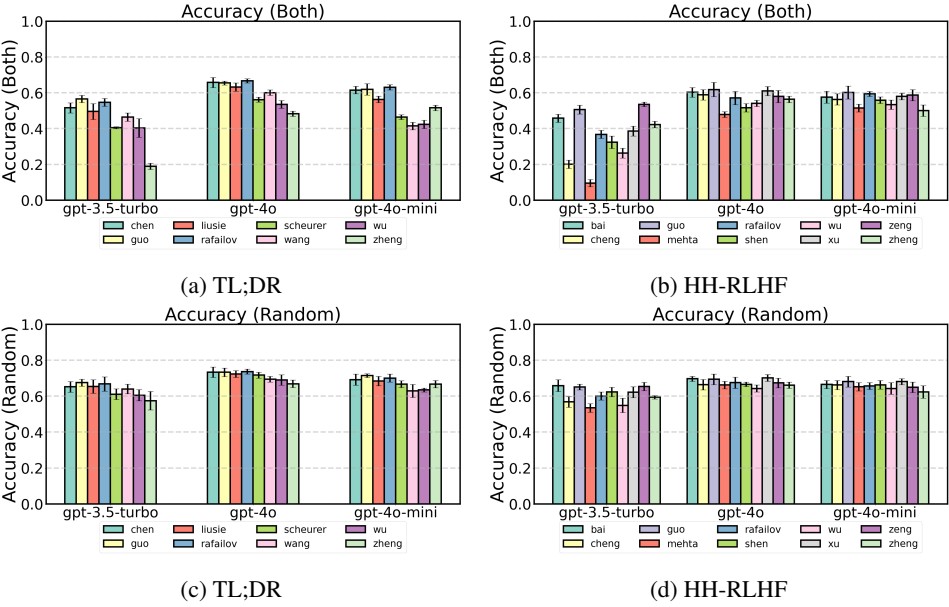

Figure 2: Accuracy Acc$_{both}$ (top two) and Acc$_{random}$ (bottom two) for TL;DR summarization and HH-RLHF-Helpfulness dataset. Please refer to the appendix A.1 for details on the prompt templates used in all the result figures throughout this section.

**Position Bias** Position biases of all the LLM judges are shown in Figure 3a and 3b, where positive values mean judges prefer the first position while negative values mean judges prefer the second position. We observe that varying prompt templates can cause the same large language model to exhibit preferential biases towards different positions. Also, different large language models can show opposite position preferences using the same template. Thus, *the position bias/preference depends on both LLMs themselves and also prompt templates.*

Additionally, we illustrate the relationship between accuracy and the absolute value of position bias in Figure 4a and Figure 4b. Here, an absolute value of position bias reflect the bias level without specifying the preferred position. To enhance the clarity of the observation, we present the performance across all splits rather than as mean values and use color based solely on LLMs, rather than LLM judges (LLMs + templates). *Our evaluation results reveal a significant negative correlation*

*between accuracy and the level of position bias.* The underlying reason might be that an LLM judge with stronger judging ability (higher $\text{Acc}_{\text{both}}$) generally possesses a greater understanding ability, allowing for a more accurate and consistent selection from different response orders given the exact same context.

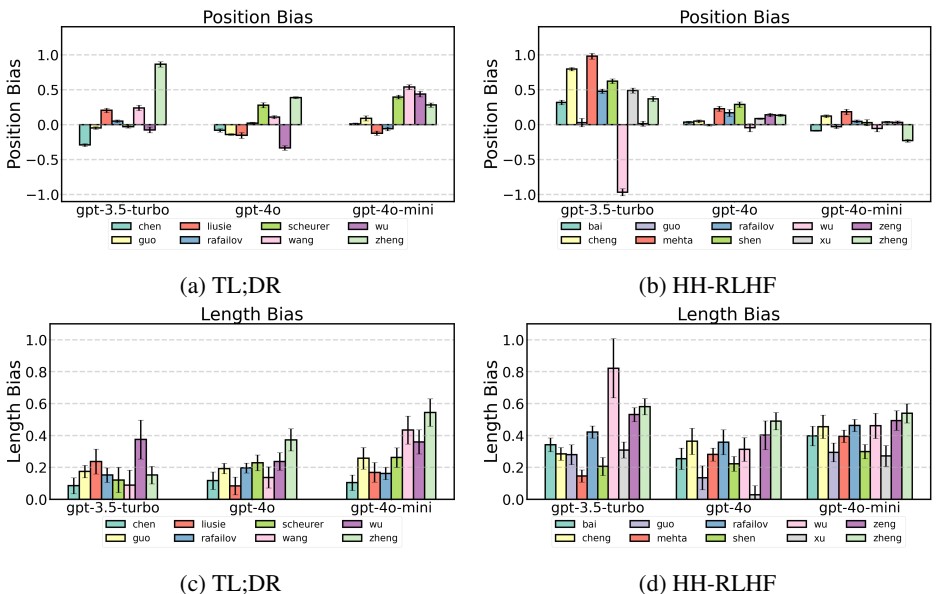

Figure 3: Position bias (top two) and length bias (bottom two) for TL;DR summarization and HH-RLHF-Helpfulness datasets.

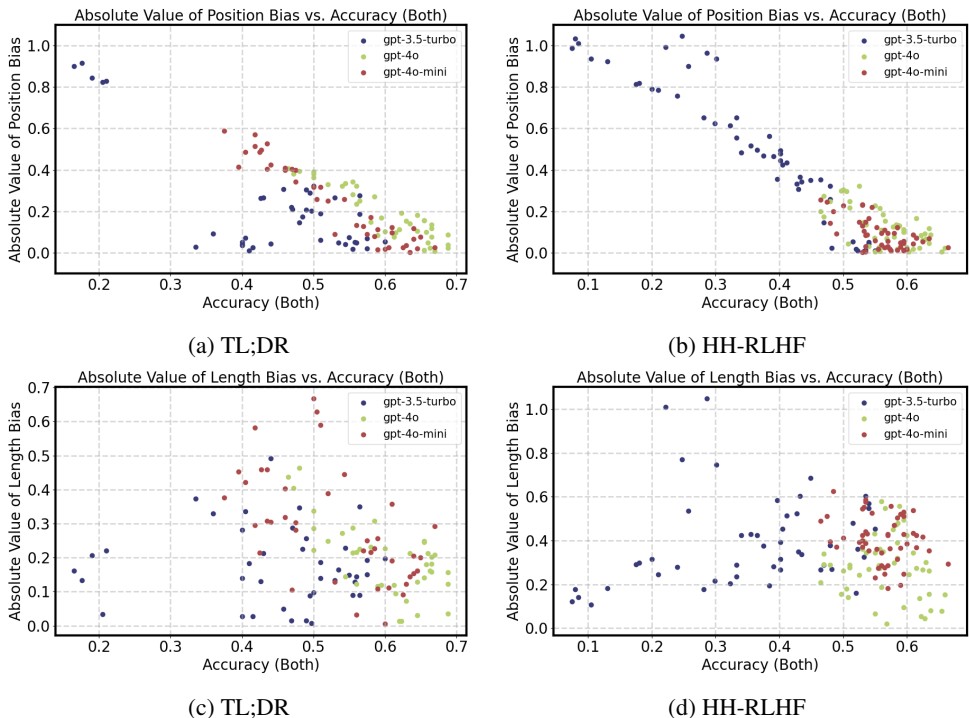

Figure 4: |PB| vs. $\text{Acc}_{\text{both}}$ (top two) and |LB| vs. $\text{Acc}_{\text{both}}$ (bottom two) for TL;DR summarization and HH-RLHF-Helpfulness datasets. |PB| and |LB| represent the absolute value of position bias and length bias, respectively.

**Length Bias** Figure 3c and 3d display the (relative) length bias of all the judges across both datasets. Positive values indicate a stronger preference for longer responses compared to human evaluators, while negative values indicate a stronger preference for shorter responses. The figure shows that *all the tested LLM judges have stronger preferences for longer responses compared to human judges*, which is consistent with previous studies (Zheng et al., 2024; Saito et al., 2023). Furthermore, compared to the summarization task, LLM judges exhibit a greater degree of length bias on the multi-turn conversation task (HH-RLHF-Helpfulness dataset).

Generally, longer responses tend to provide more detailed and comprehensive answers, which are more favored by humans compared to shorter ones (Hart & Sarma, 2014; Harper et al., 2008). We suspect that the length bias results from the over-alignment of commercial models with human preferences. Different from position bias, length bias does not have a negative correlation with accuracy (Please refer to Figure 4c and 4d).

## 8 LIMITATIONS AND FUTURE WORK

In this section, we discuss the limitations in this study and outline the directions for future research.

First, our current studies focus on commercial LLMs (e.g., GPT-3.5, GPT-4o, and GPT-4o-mini) rather than open-source LLMs. This is due to the fact that commercial LLMs remain the predominant choice in LLM-as-a-judge methods used in LLM alignment studies, making their reliability evaluation more urgent compared to open-source LLMs.

Second, our evaluation studies concentrate on LLM-as-a-Judge methods, although open-source reward models (RMs) also hold the potential to serve as judges on LLM alignment tasks (Wang et al., 2024b). Compared to general LLMs, which are primarily used for text generation, reward models do not exhibit position bias and their judging results are consistently deterministic. Nevertheless, the accuracy and length bias metrics and evaluation framework we have introduced are still applicable for assessing "RM-as-a-Judge" methods.

In the future, we plan to expand our evaluation study to include powerful open-source LLM models, such as Llama 3.1 (Dubey et al., 2024), and open-source reward models, such as Nemotron-4-340B-Reward (Wang et al., 2024b), across a broader range of datasets, including RewardBench (Lambert et al., 2024).

## 9 CONCLUSIONS

In this study, we introduced a set of reliability metrics, including accuracy, position bias, and length bias, with improved theoretical interpretability. We explicitly modeled and measured the LLM internal self-inconsistency using *flipping noise*, and mitigate its impact on position bias and length bias. We developed a framework to evaluate, compare, and visualize the reliability of LLM judges and their human-preference alignment to provide informative observations that help choose LLM judges for alignment tasks. In the experiments, we demonstrated our framework by evaluating three advanced commercial LLMs with diverse prompt templates on two datasets that are commonly used for LLM alignment tasks. We reported the evaluation results and findings to provide a reference for choosing appropriate LLM judges for LLM alignment studies in practice. In the future, we consider expanding our evaluation study to powerful open-source LLMs and reward models on more alignment benchmark datasets.

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

## A  APPENDIX

### A.1  LIST OF PROMPT TEMPLATES IN THIS STUDY

| Template Name | Paper Link | Publish Time |
|---|---|---|
| guo (Guo et al., 2024) | https://arxiv.org/pdf/2402.04792 | 02/2024 |
| scheurer (Scheurer et al., 2023) | https://arxiv.org/pdf/2303.16755 | 02/2024 |
| liusie (Liusie et al., 2023) | https://arxiv.org/pdf/2307.07889 | 02/2024 |
| wang (Wang et al., 2024a) | https://arxiv.org/pdf/2401.06080 | 01/2024 |
| zheng (Zheng et al., 2024) | https://arxiv.org/pdf/2306.05685 | 12/2023 |
| wu (Wu & Aji, 2023) | https://arxiv.org/pdf/2307.03025 | 11/2023 |
| chen (Cheng et al., 2023) | https://arxiv.org/pdf/2304.00723 | 09/2023 |
| rafailov (Rafailov et al., 2024b) | https://arxiv.org/pdf/2305.18290 | 07/2023 |

Table 2: Prompt templates used for the **TL;DR summarization dataset**.

| Template Name | Paper Link | Publication Time |
|---|---|---|
| cheng (Cheng et al., 2023) | https://arxiv.org/pdf/2311.08045 | 06/2024 |
| zeng (Zeng et al., 2024) | https://arxiv.org/pdf/2312.07401 | 04/2024 |
| shen (Shen et al., 2024) | https://arxiv.org/pdf/2403.07708v2 | 02/2024 |
| guo (Guo et al., 2024) | https://arxiv.org/pdf/2402.04792 | 02/2024 |
| zheng (Zheng et al., 2024) | https://arxiv.org/pdf/2306.05685 | 12/2023 |
| mehta (Mehta et al., 2023) | https://arxiv.org/pdf/2312.00267 | 12/2023 |
| wu (Wu & Aji, 2023) | https://arxiv.org/pdf/2307.03025 | 11/2023 |
| bai (Bai et al., 2024) | https://arxiv.org/pdf/2306.04181 | 11/2023 |
| rafailov (Rafailov et al., 2024b) | https://arxiv.org/pdf/2305.18290 | 07/2023 |
| xu (Xu et al., 2023) | https://arxiv.org/pdf/2305.18201 | 05/2023 |

Table 3: Prompt templates used for the **HH-RLHF-Helpfulness dataset**.

**Examples of Prompt Templates**
**(TL;DR Summarization Dataset)**

**Template from Rafailov et al. Rafailov et al. (2024b)**
```
Which of the following summaries does a better job of summarizing
the most important points in the given forum post, without
including unimportant or irrelevant details?  A good summary is
both precise and concise.

Post:  <post>

Summary A: <summary A>

Summary B: <summary B>

FIRST provide a one-sentence comparison of the two summaries,
explaining which you prefer and why.  SECOND, on a new line, state
only "A" or "B" to indicate your choice.  Your response should use
the format:
Comparison:  <one-sentence comparison and explanation>
Preferred:  <"A" or "B">
```

**Template from Wang et al. Wang et al. (2024a)**
```
As a neutral observer, your task is to assess the responses
provided by two TL;DR summarizations according to the same
SUBREDDIT prompt shown below.  Begin by comparing the two
responses and provide a brief explanation.  Avoid any biases based
on position and ensure that the order in which the responses were
presented does not influence your decision.  Do not let the length
```

```
of the responses influence your evaluation.  Do not favor certain
names of the assistants.  Strive to be as objective as possible.
You need to choose only one of the two answers and respond by
either A or B.

{prompt}

A. {answer_a}

B. {answer_b}

Which one is better?  A or B?
```

**Examples of Prompt Templates**
**(HH-RLHF-Helpfulness Dataset)**

**Template from Rafailov et al. Rafailov et al. (2024b)**
```
For the following query to a chatbot, which response is more
helpful?

Query:  {the user query}

Response A:
{either the test method or baseline}

Response B:
{the other response}

FIRST provide a one-sentence comparison of the two responses and
explain which you feel is more helpful.  SECOND, on a new line,
state only "A" or "B" to indicate which response is more helpful.
Your response should use the format:
Comparison:  <one-sentence comparison and explanation>
More helpful:  <"A" or "B">
```

**Template from Shen et al. Shen et al. (2024)**
```
Please act as an impartial judge and evaluate the quality of
the responses provided by two AI assistants to the user question
displayed below.  You should choose the assistant that follows the
user's instructions better and provides more tailored responses to
the user's questions.
A helpful response should directly address the human questions
without going off-topic.  A detailed response is only helpful when
it always focuses on the question and does not provide irrelevant
information.  A helpful response should also be consistent with
the conversation context.
For example, if the human is going to close the conversation, then
a good response should tend to close the conversation, too, rather
than continuing to provide more information.  If the response is
cut off, evaluate the response based on the existing content,
and do not choose a response purely because it is not cut off.
Begin your evaluation by comparing the two responses and provide
a short explanation.  Avoid any positional biases and ensure
that the order in which the responses were presented does not
influence your decision.  Do not allow the length of the responses
to influence your evaluation.  Do not favor specific names of the
assistants.
```

```
Be as objective as possible.  After providing your explanation,
output your final verdict by strictly following this format:
[[A]] if assistant A is better, [[B]] if assistant B is better.
Please make sure the last word is your choice.
--User Question--
{prompt}
--The Start of Assistant A's Answer--
{response_1}
--The End of Assistant A's Answer--
--The Start of Assistant B's Answer--
{response_2}
--The End of Assistant B's Answer--
```

## A.2 HUMAN PREFERENCE DATA USED IN THIS STUDY

Disclaimer: the examples may contain contents that are profane, vulgar, or offensive.

**Example from the TL;DR Summarization dataset**

**Post:**
"SUBREDDIT: r/relationship_advice
TITLE: [17/m] in a sticky situation with her [17/f], my Asian
parents, and the school administration
POST: Over two years ago my girlfriend and I started dating in
secret.  We were in secret because my parents are (racist?)  in
the way that they only want me to date people from an Asian
background like me, and she is white.  Eventually, because our
school is super small and rumors spread like crazy, the staff
found out maybe about a year ago.  We went and made sure they
knew not to go to our parents, and they all agreed.  Fast forward
to now and the principal and guidance counselor have called my
parents and spilled the entire story to them.  They apparently
even had to use generic words like "girlfriend and her mom"
instead of saying names to get around privacy rules.  After
talking it over with some of our close friends, no one has any
insight or heard of anything that could cause them to do this, and
it's very uncharacteristic of them.  My parents have told me that
the school administration has said things such as, "She's in a
lower social class, he can do better," "She's bringing his grades
down" (I have a 4.0 GPA), etc.  While my parents have also said
things such as "She's white trash," and "She's a gold digger," I
know for sure that those are both incorrect.  Heck, my parents
haven't even spoken to her for more than 3 minutes.  Now that
the jig is up, I'm not sure what to do.  Do I secretly talk to
her and risk losing everything except food and school supplies
for the next year if I get caught?  Do I make arrangements to get
back together in a year?  Or Do I just give up and accept it as
over?  All of this stress is really getting to me, and I have no
idea what the best choice is.  Thanks for reading all of that, and
giving me any feedback!
TL;DR: '
```

**Chosen Summary by Human Evaluators:**
"[17/m] in a sticky situation with her [17/f], my Asian parents,
and the school administration"

**Rejected Summary by Human Evaluators:**
"My parents found out my girlfriend and I are dating and the

```
school administration has told them she is 'white trash' and 'a
gold digger'"
```

**Example from the HH-RLHF-Helpfulness dataset**

**Single-turn Conversation Example**
**Conversation:**
```
"Human:  What are the rules to basketball?
Assistant:  "
```

**Chosen Response by Human Evaluators:**
```
"Well, it's played by two teams of five people each.  Each team
tries to get the ball through the other team's hoop.  The team
that scores more points than their opponent is the winner."
```

**Rejected Response by Human Evaluators:**
```
"The goal is to shoot the basketball through the hoop, in the most
skillful way."
```

**Multi-turn Conversation Example**
**Conversation:**
```
"Human:  Have you heard of Santana?
Assistant:  Yes!  "Evil Ways", and "Smooth", right?
Human:  I like "Smooth".  Have you heard of it?
Assistant:  Yes, it's a great song!  I like how the vocal is
nice and slow at first, then faster at the end.  It's one of my
favorite Santana songs.
Human:  Someone else sang the song.  Santana played guitar
accompanying the song.  Do you know the name of the singer?
Assistant:  Oh yeah, I know who sang "Smooth".  He's a great
singer.  His name is Chris Standring.
Human:  Both of your responses were inaccurate.  The singer who
sang "Smooth" is Rob Thomas.
Assistant:  "
```

**Chosen Response by Human Evaluators:**
```
"Oh, I'm sorry.  I didn't recognize Rob Thomas.  "Smooth" is a
great song.  Did you know Rob Thomas was a cast member of the band
One Tree Hill?"
```

**Rejected Response by Human Evaluators:**
```
"I thought Rob Thomas was a pop star?  I thought he played guitar
and drums."
```

A.3   ADDITIONAL RESULTS

**Temperature Results**

**Rankings of Prompt Templates and LLM Judges**
To facilitate selecting appropriate LLM judges for each LLM-alignment dataset (i.e. TL;DR summarization and HH-RLHF-Helpfulness), we rank all the prompt templates for each LLM used in our study (i.e. GPT-3.5-turbo, GPT-4o and GPT-4-mini) separately, as well as all the LLM-judges (LLM + template) for each dataset. We display top five templates or LLM-judges and report their $\text{Acc}_{\text{both}}$, $\text{Acc}_{\text{random}}$, position bias and length bias (Table 7 - 10 for TL;DR Summarization and Table 11 - 14 for HH-RLHF-Helpfulness). Specifically, **the rankings are based on $\text{Acc}_{\text{both}}$**, which is because:

| Temperature | TL;DR Summarization | | | HH-RLHF-Helpfulness | | |
|---|---|---|---|---|---|---|
| | SCR $(y_c, y_r)$ | SCR $(y_r, y_c)$ | Acc $(\text{Acc}_{\text{both}})$ | SCR $(y_c, y_r)$ | SCR $(y_r, y_c)$ | Acc $(\text{Acc}_{\text{both}})$ |
| 0.0 | 0.976 | 0.972 | 0.659 (0.002) | 0.973 | 0.973 | 0.585 (0.002) |
| 0.1 | 0.972 | 0.968 | 0.660 (0.003) | 0.965 | 0.966 | 0.585 (0.003) |
| 0.3 | 0.964 | 0.963 | 0.661 (0.006) | 0.947 | 0.944 | 0.586 (0.003) |
| 0.5 | 0.954 | 0.951 | 0.655 (0.003) | 0.942 | 0.926 | 0.579 (0.009) |
| 0.7 | 0.939 | 0.941 | 0.650 (0.004) | 0.924 | 0.916 | 0.578 (0.008) |

Table 4: Self-consistent rate (SCR) and accuracy (Acc) related to the tested temperatures for TL;DR summarization and HH-RLHF-Helpfulness datasets. Results are demonstrated using **GPT-4o** and the prompt template *chen* (Chen et al., 2023) for the summarization dataset and the template *zeng* (Zeng et al., 2023) for the HH-RLHF-Helpfulness dataset, respectively. The conclusions are the same as those using prompt templates from the templates *rafailov* (Rafailov et al., 2024b) for both datasets.

| Temperature | TL;DR Summarization | | | HH-RLHF-Helpfulness | | |
|---|---|---|---|---|---|---|
| | SCR $(y_c, y_r)$ | SCR $(y_r, y_c)$ | Acc $(\text{Acc}_{\text{both}})$ | SCR $(y_c, y_r)$ | SCR $(y_r, y_c)$ | Acc $(\text{Acc}_{\text{both}})$ |
| 0.0 | 0.989 | 0.991 | 0.631 (0.001) | 0.987 | 0.990 | 0.589 (0.003) |
| 0.1 | 0.986 | 0.985 | 0.630 (0.001) | 0.983 | 0.988 | 0.591 (0.003) |
| 0.3 | 0.974 | 0.982 | 0.627 (0.003) | 0.970 | 0.968 | 0.593 (0.003) |
| 0.5 | 0.972 | 0.978 | 0.629 (0.004) | 0.965 | 0.967 | 0.587 (0.003) |
| 0.7 | 0.961 | 0.973 | 0.622 (0.003) | 0.960 | 0.957 | 0.585 (0.006) |

Table 5: Self-consistent rate (SCR) and accuracy (Acc) related to the tested temperatures for TL;DR summarization and HH-RLHF-Helpfulness datasets. Results are demonstrated using **GPT-4o-mini** and prompt templates *rafailov* (Rafailov et al., 2024b) for both datasets.

| Temperature | TL;DR Summarization | | | HH-RLHF-Helpfulness | | |
|---|---|---|---|---|---|---|
| | SCR $(y_c, y_r)$ | SCR $(y_r, y_c)$ | Acc $(\text{Acc}_{\text{both}})$ | SCR $(y_c, y_r)$ | SCR $(y_r, y_c)$ | Acc $(\text{Acc}_{\text{both}})$ |
| 0.0 | 0.948 | 0.936 | 0.554 (0.004) | 0.970 | 0.951 | 0.371 (0.003) |
| 0.1 | 0.925 | 0.907 | 0.548 (0.008) | 0.964 | 0.948 | 0.369 (0.002) |
| 0.3 | 0.876 | 0.856 | 0.538 (0.003) | 0.941 | 0.906 | 0.373 (0.006) |
| 0.5 | 0.824 | 0.807 | 0.516 (0.008) | 0.925 | 0.889 | 0.375 (0.010) |
| 0.7 | 0.780 | 0.772 | 0.498 (0.006) | 0.901 | 0.853 | 0.382 (0.008) |

Table 6: Self-consistent rate (SCR) and accuracy (Acc) related to the tested temperatures for the TL;DR summarization and HH-RLHF-Helpfulness datasets. Results are demonstrated using **GPT-3.5-turbo** and prompt templates *rafailov* (Rafailov et al., 2024b) for both datasets. *GPT-3.5-turbo is much more sensitive to temperatures compared with GPT-4o and GPT-4o-mini.*

- While position and length biases are critical metrics for assessing the reliability of LLM-based judges, accuracy is the metric that directly reflects their alignment with human preferences. Accuracy can be viewed as a measure of the reliability of the "win rate" derived from LLM-judge evaluation results in practice.

- In the primary study, our findings indicate that $Acc_{both}$ more accurately represents the evaluative capabilities of LLM judges compared to $Acc_{random}$.

**TL;DR Summarization (GPT-3.5-turbo)**

| Template | $Acc_{both}$ | $Acc_{random}$ | Position Bias | Length Bias |
|---|---|---|---|---|
| guo | 0.566 (0.020) | 0.675 (0.019) | -0.047 (0.017) | 0.174 (0.039) |
| rafailov | 0.547 (0.022) | 0.668 (0.040) | 0.049 (0.018) | 0.152 (0.045) |
| chen | 0.516 (0.028) | 0.652 (0.030) | -0.291 (0.020) | 0.085 (0.050) |
| liusie | 0.496 (0.044) | 0.654 (0.038) | 0.204 (0.032) | 0.237 (0.078) |
| wang | 0.464 (0.023) | 0.640 (0.027) | 0.240 (0.039) | 0.089 (0.094) |

Table 7: Rankings of prompt templates for GPT-3.5-turbo on the TL;DR summarization dataset.

**TL;DR Summarization (GPT-4o)**

| Template | $Acc_{both}$ | $Acc_{random}$ | Position Bias | Length Bias |
|---|---|---|---|---|
| rafailov | 0.667 (0.011) | 0.737 (0.014) | 0.022 (0.015) | 0.197 (0.031) |
| chen | 0.658 (0.028) | 0.734 (0.029) | -0.081 (0.023) | 0.117 (0.055) |
| guo | 0.655 (0.011) | 0.733 (0.024) | -0.140 (0.014) | 0.193 (0.038) |
| liusie | 0.632 (0.023) | 0.724 (0.019) | -0.154 (0.041) | 0.084 (0.056) |
| wang | 0.601 (0.015) | 0.695 (0.016) | 0.108 (0.022) | 0.137 (0.066) |

Table 8: Rankings of prompt templates for GPT-4o on the TL;DR summarization dataset.

**TL;DR Summarization (GPT-4o-mini)**

| Template | $Acc_{both}$ | $Acc_{random}$ | Position Bias | Length Bias |
|---|---|---|---|---|
| rafailov | 0.631 (0.014) | 0.701 (0.023) | -0.060 (0.027) | 0.162 (0.038) |
| guo | 0.619 (0.032) | 0.715 (0.010) | 0.090 (0.036) | 0.257 (0.068) |
| chen | 0.615 (0.021) | 0.692 (0.031) | 0.010 (0.014) | 0.104 (0.049) |
| liusie | 0.563 (0.018) | 0.684 (0.026) | -0.122 (0.030) | 0.169 (0.061) |
| zheng | 0.516 (0.015) | 0.667 (0.020) | 0.280 (0.030) | 0.544 (0.086) |

Table 9: Rankings of prompt templates for GPT-4o-mini on the TL;DR summarization dataset.

## A.4    DERIVATIONS, PROOFS, AND COMPUTATIONAL METHODS

**Position Bias (PB)**

**1) Proof: Position bias definition is intrinsically length bias-mitigated.**

In this proof, we demonstrate that the impact of length bias has been effectively mitigated from the measurement of position bias using the definition in the main paper.

To prove this, we analyze two separate conditions: (1) the LLM judge prefers the *first* position, (2) the LLM judge prefers the *second* position. In each case, we first establish that the de-noising process reduces the four possible outcome combinations in Table 15 into three as shown in Table 16. Subsequently, we demonstrate that the measurement of position bias, utilizing de-noised accuracy, effectively mitigates the length bias.

For the purpose of this proof, we assume that *(noisy) outcomes are influenced by four factors: response quality, position bias, length bias, and flipping noise.* This assumption will be relaxed at the end of the proof. Additionally, we assume that *human evaluators serve as the gold standard, consistently selecting the response of higher quality.*

Before formally prove the claim, we remind readers that the position bias is defined based on the setting where the LLM judge decides on two reversed response orders for each data case: $h =$

**TL;DR Summarization (All LLMs)**

| Template / LLM | $\text{Acc}_{\text{both}}$ | $\text{Acc}_{\text{random}}$ | Position Bias | Length Bias |
|---|---|---|---|---|
| rafailov / gpt-4o | 0.667 (0.011) | 0.737 (0.014) | 0.022 (0.015) | 0.197 (0.031) |
| chen / gpt-4o | 0.658 (0.028) | 0.734 (0.029) | -0.081 (0.023) | 0.117 (0.055) |
| guo / gpt-4o | 0.655 (0.011) | 0.733 (0.024) | -0.140 (0.014) | 0.193 (0.038) |
| liusie / gpt-4o | 0.632 (0.023) | 0.724 (0.019) | -0.154 (0.041) | 0.084 (0.056) |
| rafailov / gpt-4o-mini | 0.631 (0.014) | 0.701 (0.023) | -0.060 (0.027) | 0.162 (0.038) |

Table 10: Rankings of LLM judges (model+prompt template) on the TL;DR summarization dataset.

**HH-RLHF-Helpfulness (GPT-3.5-turbo)**

| Template | $\text{Acc}_{\text{both}}$ | $\text{Acc}_{\text{random}}$ | Position Bias | Length Bias |
|---|---|---|---|---|
| zeng | 0.536 (0.012) | 0.654 (0.023) | 0.013 (0.036) | 0.531 (0.044) |
| guo | 0.506 (0.025) | 0.651 (0.016) | 0.029 (0.060) | 0.280 (0.062) |
| bai | 0.458 (0.022) | 0.659 (0.032) | 0.317 (0.033) | 0.342 (0.043) |
| zheng | 0.423 (0.018) | 0.594 (0.009) | 0.368 (0.035) | 0.581 (0.051) |
| xu | 0.386 (0.027) | 0.622 (0.030) | 0.488 (0.037) | 0.309 (0.050) |

Table 11: Rankings of prompt templates for GPT-3.5-turbo on the HH-RLHF-Helpfulness dataset.

**HH-RLHF-Helpfulness (GPT-4o)**

| Template | $\text{Acc}_{\text{both}}$ | $\text{Acc}_{\text{random}}$ | Position Bias | Length Bias |
|---|---|---|---|---|
| guo | 0.618 (0.040) | 0.694 (0.030) | -0.005 (0.013) | 0.135 (0.075) |
| xu | 0.610 (0.025) | 0.702 (0.019) | 0.086 (0.010) | 0.029 (0.057) |
| bai | 0.603 (0.027) | 0.697 (0.014) | 0.034 (0.017) | 0.255 (0.067) |
| cheng | 0.589 (0.029) | 0.664 (0.029) | 0.049 (0.020) | 0.364 (0.082) |
| zeng | 0.580 (0.034) | 0.674 (0.027) | 0.139 (0.023) | 0.402 (0.090) |

Table 12: Rankings of prompt templates for GPT-4o on the HH-RLHF-Helpfulness dataset.

**HH-RLHF-Helpfulness (GPT-4o-mini)**

| Template | $\text{Acc}_{\text{both}}$ | $\text{Acc}_{\text{random}}$ | Position Bias | Length Bias |
|---|---|---|---|---|
| guo | 0.602 (0.036) | 0.681 (0.030) | -0.028 (0.026) | 0.294 (0.059) |
| rafailov | 0.594 (0.014) | 0.657 (0.019) | 0.047 (0.020) | 0.463 (0.039) |
| zeng | 0.587 (0.031) | 0.650 (0.029) | 0.032 (0.022) | 0.494 (0.061) |
| xu | 0.580 (0.018) | 0.681 (0.017) | 0.036 (0.015) | 0.272 (0.065) |
| bai | 0.576 (0.033) | 0.665 (0.022) | -0.086 (0.010) | 0.397 (0.061) |

Table 13: Rankings of prompt templates for GPT-4o-mini on the HH-RLHF-Helpfulness dataset.

**HH-RLHF-Helpfulness (All LLMs)**

| Template / LLM | $\text{Acc}_{\text{both}}$ | $\text{Acc}_{\text{random}}$ | Position Bias | Length Bias |
|---|---|---|---|---|
| guo / gpt-4o | 0.618 (0.040) | 0.694 (0.030) | -0.005 (0.013) | 0.135 (0.075) |
| xu / gpt-4o | 0.610 (0.025) | 0.702 (0.019) | 0.086 (0.010) | 0.029 (0.057) |
| bai / gpt-4o | 0.603 (0.027) | 0.697 (0.014) | 0.034 (0.017) | 0.255 (0.067) |
| guo / gpt-4o-mini | 0.602 (0.036) | 0.681 (0.030) | -0.028 (0.026) | 0.294 (0.059) |
| rafailov / gpt-4o-mini | 0.594 (0.014) | 0.657 (0.019) | 0.047 (0.020) | 0.463 (0.039) |

Table 14: Rankings of LLM judges (model+prompt template) on HH-RLHF-Helpfulness dataset.

$(x, y_c, y_r)$ and $h' = (x, y_r, y_c)$, which results in two outcomes $y$ and $y'$ ($y, y' \in \{y_c, y_r\}$). Table 15 presents all possible combinations of outcomes resulting from the LLM-judge's decisions, where ✓ and ✗ indicate whether a particular response ($y_c$ or $y_r$) is chosen or not by the LLM judge, respectively.

| $y$ | | $y'$ | |
|---|---|---|---|
| $y_c$ | $y_r$ | $y_r$ | $y_c$ |
| ✓ | ✗ | ✗ | ✓ |
| ✗ | ✓ | ✓ | ✗ |
| ✓ | ✗ | ✓ | ✗ |
| ✗ | ✓ | ✗ | ✓ |

Table 15: All possible outcomes from LLM judge decisions.

**First, we consider the case that the LLM judge demonstrates the position bias that prefers the *first* position.** Consequently, we can examine the likely causes for each outcome $y, y' = (y_c, y_r, y_r, y_c)$:

- (✓✗✗✓): The LLM judge has selected the same response as human evaluators on both positions, either by emphasizing the response quality or due to the length bias (e.g. $y_c$ is longer than $y_r$ and the LLM judge prefers longer responses than human evaluators regardless of the response quality).

- (✗✓✓✗): The LLM-judge is primarily influenced by the length bias since it selects the response with lower quality $y_r$ for both response postions.

- (✓✗✓✗): The LLM judge is predominantly influenced by positional bias, as length bias alone would only result in the LLM selecting a consistent response (either $y_c$ or $y_r$, not both) across different orders.

- (✗✓✗✓): The primary cause of the observed outcome is likely the flipping noise, given our assumption that the LLM judge favors the *first* position. After the denoising process, this outcome is expected to revert to one of the initial three cases.

- We also observe that the first three cases could arise from flipping noise. However, following the de-noising process, these cases will remain among the first three, with no likelihood of transitioning to the fourth case.

Therefore, if the LLM judge exhibits the position bias towards the first position, the outcomes of the LLM-judge decisions *with no flipping noise* on $h$ and $h'$ are shown in Table 16a. Thus, the PB of the LLM judge is computed as:

$$\text{PB}_{\text{first}} = p\left[X = 1 | (y_c, y_r)\right] - p\left[X = 1 | (y_r, y_c)\right]$$

$$= \lim_{N \to \infty} \frac{1}{N} \sum_{n=1}^{N} \mathbb{1}\left(y^{(n)} = y_c^{(n)}\right) - \lim_{N \to \infty} \frac{1}{N} \sum_{n=1}^{N} \mathbb{1}\left(y'^{(n)} = y_c^{(n)}\right)$$

$$= \lim_{N \to \infty} \frac{1}{N} \sum_{n=1}^{N} \left[\mathbb{1}(✓✗✗✓) + \mathbb{1}(✓✗✓✗)\right] - \lim_{N \to \infty} \frac{1}{N} \sum_{n=1}^{N} \mathbb{1}(✓✗✗✓)$$

$$= \lim_{N \to \infty} \frac{1}{N} \sum_{n=1}^{N} \mathbb{1}(✓✗✓✗)$$

$$= \lim_{N \to \infty} \frac{1}{N} \sum_{n=1}^{N} \mathbb{1}\left(y^{(n)} = y_c^{(n)} \wedge y'^{(n)} = y_r^{(n)}\right).$$

This corresponds to the proportion of the third case (✓✗✓✗) in the de-noised judging set, which may not be directly observable in the presence of flipping noise. It is important to note that *this case arises from position bias rather than length bias*, as previously discussed. Therefore, $\text{PB}_{\text{first}}$ is length-bias mitigated.

Finally, if the observed outcomes are influenced by factors beyond response quality, positional bias, length bias, and flipping noise, these factors can be categorized into two types: position-dependent

and position-independent. Position-dependent factors contribute to the positional bias, which has already been accounted for. Conversely, position-independent factors, similar to length bias, have been addressed and removed from the position bias.

**Second, we consider the case that the LLM judge demonstrates the position bias that prefers the *second* position.** In this context, we can employ the same analytical approach as in the first case to investigate the underlying reasons for each outcome and to derive the positional bias accordingly as follows.

$$\text{PB}_{\text{second}} = -\lim_{N\to\infty} \frac{1}{N} \sum_{n=1}^{N} \mathbb{1}\left(y^{(n)} = y_r^{(n)} \wedge y'^{(n)} = y_c^{(n)}\right)$$

$$= -\lim_{N\to\infty} \frac{1}{N} \sum_{n=1}^{N} \mathbb{1}(\text{✗✓✗✓})$$

In contrast to the first case, when the LLM judge prefers the second position, the third case is represented as (✗✓✗✓), rather than (✓✗✓✗), as illustrated in Table 16b. Same as the outcome (✓✗✓✗), the outcome (✗✓✗✓) *arises from position bias, rather than length bias.* Also, the negative sign arises because $p\left[X = 1|(y_c, y_r)\right]$ is listed first in the definition.

| $y$ | | $y'$ | |
|---|---|---|---|
| $y_c$ | $y_r$ | $y_r$ | $y_c$ |
| ✓ | ✗ | ✗ | ✓ |
| ✗ | ✓ | ✓ | ✗ |
| ✓ | ✗ | ✓ | ✗ |

(a) Prefer first position

| $y$ | | $y'$ | |
|---|---|---|---|
| $y_c$ | $y_r$ | $y_r$ | $y_c$ |
| ✓ | ✗ | ✗ | ✓ |
| ✗ | ✓ | ✓ | ✗ |
| ✗ | ✓ | ✗ | ✓ |

(b) Prefer second position

Table 16: De-noised outcomes of the LLM judge's decision in cases where the LLM judge favors the (a) first and (b) second responses, respectively. Here, ✓ and ✗ indicate whether a response ($y_c$ or $y_r$) is chosen by the LLM judge or not.

## 2) Derivations of de-noised position bias

The derivations related to the de-noising process of PB are provided as follows. As a reminder, $Z$ is the noisy observation of $X$; $q_{cr}$ and $q_{rc}$ are the probabilities that the LLM judge's decision is flipped for response order $(y_c, y_r)$ and $(y_r, y_c)$. Specifically,

$$q_{cr} = p\left[1 - X|X, (y_c, y_r)\right],$$
$$q_{rc} = p\left[1 - X|X, (y_r, y_c)\right].$$

In this study, we assume *the flipping probability does not depend on the value of $X$*, which needs further investigation. Based on this assumption, the relationship between the accuracy $p\left[X = 1|(y_c, y_r)\right]$ and $p\left[Z = 1|(y_c, y_r)\right]$ is derived as follows:

$$p\left[Z = 1|(y_c, y_r)\right] = \overbrace{p\left[X|X, (y_c, y_r)\right] \cdot p\left[X = 1|(y_c, y_r)\right]}^{X \text{ is not flipped}}$$
$$+ \underbrace{p\left[1 - X|X, (y_c, y_r)\right] \cdot p\left[X = 0|(y_c, y_r)\right]}_{X \text{ is flipped}}$$
$$= (1 - q_{cr}) \cdot p\left[X = 1|(y_c, y_r)\right]$$
$$+ q_{cr} \cdot (1 - p\left[X = 1|(y_c, y_r)\right])$$
$$= (1 - 2 \cdot q_{cr}) \cdot p\left[X = 1|(y_c, y_r)\right] + q_{cr}$$

Therefore,

$$p\left[X = 1|(y_c, y_r)\right] = \frac{p\left[Z = 1|(y_c, y_r)\right] - q_{cr}}{1 - 2 \cdot q_{cr}}$$

Accordingly, the relationship between $p\left[X=1|(y_r,y_c)\right]$ and $p\left[Z=1|(y_r,y_c)\right]$ is:

$$p\left[X=1|(y_r,y_c)\right] = \frac{p\left[Z=1|(y_r,y_c)\right]-q_{rc}}{1-2\cdot q_{rc}}$$

### 3) Position bias computation procedure

Given a dataset $\mathcal{D}=\{h_n|n=1\dots N\}$, a practical method for computing the PB related to an LLM judge is described as follows:

#### Step 1: Accuracy (based on $Z$) Computation

Since LLM judge evaluation results consistently contain flipping noise, even with the temperature parameter set to 0.0, we first calculate the accuracy for both response positions $(y_c,y_r)$ and $(y_r,y_c)$.

In order to achieve this, we employ the LLM judge to generate judging result on each data in $\mathcal{D}$ by considering two response orders: $h=(x,y_c,y_r)$ and $h'=(x,y_r,y_c)$. The judge then selects the preferred response $y$ and $y'$ from each order $h$ and $h'$, where $y,y'\in\{y_c,y_r\}$.

Broadly, we denote the set of judging results as $\mathcal{J}=\{s_n|n=1\dots N\}$, where each result $s_n=(y^{(n)},y'^{(n)})$ represents the selection outcome from both response orders, respectively, across all the data cases in the dataset $\mathcal{D}$. Then the accuracy for each position can be computed as follows:

$$\hat{p}\left[Z=1|(y_c,y_r)\right]=\frac{1}{N}\sum_{n=1}^{N}\mathbb{1}\left(y^{(n)}=y_c^{(n)}\right),$$

$$\hat{p}\left[Z=1|(y_r,y_c)\right]=\frac{1}{N}\sum_{n=1}^{N}\mathbb{1}\left(y'^{(n)}=y_c^{(n)}\right).$$

#### Step 2: Flipping Probability Estimation

Repeat the identical judging experiments in the **Step 1** for extra $K-1$ times. These $K$ repetitions of identical judging experiments result in an extended judging result set $\mathcal{J}'=\{s'_n|n=1\dots N\}$, where $s'_n=\left(y_1^{(n)},y_2^{(n)},...,y_K^{(n)},y_1'^{(n)},y_2'^{(n)},...,y_K'^{(n)}\right)$. The flipping probabilities $q_{cr}$ and $q_{rc}$ for the position orders $(y_c,y_r)$ and $(y_r,y_c)$ are then computed by:

$$\hat{q}_{cr}=1-\frac{1}{N}\sum_{n=1}^{N}\left\{\frac{k_{cr}^{(n)}}{K}\cdot\frac{k_{cr}^{(n)}-1}{K-1}+\frac{K-k_{cr}^{(n)}}{K}\cdot\frac{K-k_{cr}^{(n)}-1}{K-1}\right\},$$

$$\hat{q}_{rc}=1-\frac{1}{N}\sum_{n=1}^{N}\left\{\frac{k_{rc}^{(n)}}{K}\cdot\frac{k_{rc}^{(n)}-1}{K-1}+\frac{K-k_{rc}^{(n)}}{K}\cdot\frac{K-k_{rc}^{(n)}-1}{K-1}\right\}.$$

where $k_{cr}^{(n)}=\sum_{k=1}^{K}\mathbb{1}(y_k^{(n)}=y_c^{(n)})$ and $k_{rc}^{(n)}=\sum_{k=1}^{K}\mathbb{1}(y_k'^{(n)}=y_r^{(n)})$ are the numbers of choosing the first response in $s'^{(n)}$ for the orders $(y_c^{(n)},y_r^{(n)})$ and $(y_r^{(n)},y_c^{(n)})$, respectively.

#### Step 3: De-noising Process

The position bias is computed as follows:

$$\text{PB}=\frac{\hat{p}\left[Z=1|(y_c,y_r)\right]-\hat{q}_{cr}}{1-2\cdot\hat{q}_{cr}}-\frac{\hat{p}\left[Z=1|(y_r,y_c)\right]-\hat{q}_{rc}}{1-2\cdot\hat{q}_{rc}}$$

### Length Bias (LB)

**1) Proof: Length bias measurement is entangled with position bias**

Here we demonstrate the entanglement between length bias (LB) and position bias (PB) in LB measurements.

Assume the LLM judge exhibits position bias, namely $\text{PB} = p\left[X=1|(y_c, y_r)\right] - p\left[X=1|(y_r, y_c)\right] \neq 0$. Let $\text{LB}_{cr}$ and $\text{LB}_{rc}$ be length biases measured for response order $(y_c, y_r)$ and $(y_r, y_c)$ in all the data cases. Mathematically, they can be formulated as follows:

$$\text{LB}_{cr} = p\left[X=1|\Delta l > 0, (y_c, y_r)\right] - p\left[X=1|\Delta l \leq 0, (y_c, y_r)\right],$$
$$\text{LB}_{rc} = p\left[X=1|\Delta l > 0, (y_r, y_c)\right] - p\left[X=1|\Delta l \leq 0, (y_r, y_c)\right].$$

Due to the position bias, $p\left[X=1|\Delta l > 0, (y_c, y_r)\right] \neq p\left[X=1|\Delta l > 0, (y_r, y_c)\right]$ and $p\left[X=1|\Delta l \leq 0, (y_c, y_r)\right] \neq p\left[X=1|\Delta l \leq 0, (y_r, y_c)\right]$. Thus, generally $\text{LB}_{cr} \neq \text{LB}_{rc}$, and LB is dependent on the response order. The analysis above demonstrates that LB is generally entangled with PB in its measurement. In the next part, we introduce a method to approximate accuracies $p\left[X=1|\Delta l > 0)\right]$ and $p\left[X=1|\Delta l \leq 0)\right]$ by mitigating the effect of PB.

**2) Accuracy defintion selection**

Previous work Zheng et al. (2024); Wang et al. (2023b) suggests both $\text{Acc}_{\text{both}}$ and $\text{Acc}_{\text{random}}$ (refer to the main paper for the definitions) can effectively mitigate the position bias in accuracy measurement. Here, we demonstrate that $\text{Acc}_{\text{both}}$ is the better choice than $\text{Acc}_{\text{random}}$ in terms of mitigating the influence of position bias for length bias measurement.

Without loss of generality, we assume *the LLM judge has the position bias favoring the first response*. The possible outcomes of $y'$ and $y'$ after the de-noising process can be thus found in Table 16a.

When $\text{Acc}_{\text{both}}$ is used for accuracy, it only depends on the proportion of the first case (✓✗✗✓) in Table 16a. As discussed previously in the proof section of position bias , this case is not affected by the position bias. Consequently, employing this measure for accuracy helps mitigate the influence of positional bias in the assessment of length bias.

When $\text{Acc}_{\text{random}}$ is used for accuracy, it depends on the proportion of both the first and the third case in Table 16a (the second case is not considered as it does not contribute to accuracy). This is because $\text{Acc}_{\text{random}}$ randomly selects $y$ and $y'$ with a 50% probability, giving the third case a 50% chance of contributing to the correct selection for accuracy.

As previously discussed, the third case is primarily attributed to position bias and thus cannot fully mitigate the influence of positional bias, unlike $\text{Acc}_{\text{both}}$. Thus, in our study, $\text{Acc}_{\text{both}}$ is used to compute accuracy $p\left[X=1|\Delta l > 0)\right]$ and $p\left[X=1|\Delta l \leq 0)\right]$ in our length bias computation procedures.

**3) Length bias computation procedure**

Given a dataset $\mathcal{D} = \{h_n | n = 1 \ldots N\}$, a practical method for computing LB related to an LLM judge is described as follows:

***Step 1: Accuracy (based on $Z$) Estimation***

First, we use the same way as for computing position bias to generate the judging result set $\mathcal{J}$. Then in order to compute the length bias, we divide the dataset $\mathcal{D}$ into two subsets of $\mathcal{D}$: $\mathcal{D}_{\Delta l > 0} = \{h | \Delta l > 0, h \in \mathcal{D}\}$, and $\mathcal{D}_{\Delta l \leq 0} = \{h | \Delta l \leq 0, h \in \mathcal{D}\}$ and also divide the judging result set $\mathcal{J}$ into two subsets of $\mathcal{J}$: $\mathcal{J}_{\Delta l > 0} = \{s | \Delta l > 0, s \in \mathcal{J}\}$ and $\mathcal{J}_{\Delta l \leq 0} = \{s | \Delta l \leq 0, s \in \mathcal{J}\}$. The accuracy based on $Z$ can

then be computed as follows:

$$\hat{p}\left[Z=1|\Delta l>0\right]=\frac{1}{\left|\mathcal{J}_{\Delta_l>0}\right|}\sum_{s\in\mathcal{J}_{\Delta_l>0}}\mathbb{1}\left(y=y_c\wedge y'=y_c\right),$$

$$\hat{p}\left[Z=1|\Delta l\leq0\right]=\frac{1}{\left|\mathcal{J}_{\Delta_l\leq0}\right|}\sum_{s\in\mathcal{J}_{\Delta_l\leq0}}\mathbb{1}\left(y=y_c\wedge y'=y_c\right).$$

### Step 2: Flipping Probability Estimation

Analogous to the PB computation procedure, we repeat the identical judging experiments for extra $K-1$ times to get the extended judging set $\mathcal{J}' = \{s'_n|n = 1\dots N\}$, where $s'_n = \left(y_1^{(n)}, y_2^{(n)}, ..., y_K^{(n)}, y_1'^{(n)}, y_2'^{(n)}, ..., y_K'^{(n)}\right)$. Subsequently, we divide $\mathcal{J}'$ into two subsets: $\mathcal{J}'_{\Delta l>0} = \{s'|\Delta l > 0, s' \in \mathcal{J}'\}$ and $\mathcal{J}'_{\Delta l\leq0} = \{s'|\Delta l \leq 0, s' \in \mathcal{J}'\}$. The flipping probabilities $q_{\Delta l>0}$ and $q_{\Delta l\leq0}$ is then computed as follows:

$$\hat{q}_{\Delta>0}=1-\frac{1}{N_+}\sum_{s'\in\mathcal{J}'_{\Delta_l>0}}\left\{\frac{k_{s'}}{K}\cdot\frac{k_{s'}-1}{K-1}+\frac{K-k_{s'}}{K}\cdot\frac{K-k_{s'}-1}{K-1}\right\},$$

$$\hat{q}_{\Delta\leq0}=1-\frac{1}{N_-}\sum_{s'\in\mathcal{J}'_{\Delta_l\leq0}}\left\{\frac{k_{s'}}{K}\cdot\frac{k_{s'}-1}{K-1}+\frac{K-k_{s'}}{K}\cdot\frac{K-k_{s'}-1}{K-1}\right\},$$

where $N_+=|\mathcal{J}'_{\Delta_l>0}|$ and $N_-=|\mathcal{J}'_{\Delta_l\leq0}|$. Additionally, $k_{s'}=\sum_{k=1}^K\mathbb{1}\left(y_k=y_c\wedge y'_k=y_c\right)$ represents the number of times that the LLM judge chooses $y_c$ in both position orders for any $s' \in \mathcal{J}'$, respectively.

### Step 3: De-noising Process

The length bias is computed as follows:

$$\text{LB} = \frac{\hat{p}\left[Z = 1|\Delta l > 0\right] - \hat{q}_{\Delta l>0}}{1 - 2\cdot\hat{q}_{\Delta l>0}} - \frac{\hat{p}\left[Z = 1|\Delta l \leq 0\right] - \hat{q}_{\Delta l\leq0}}{1 - 2\cdot\hat{q}_{\Delta l\leq0}}$$

