# OpenReview forum: "Systematic Evaluation of LLM-as-a-Judge in LLM Alignment Tasks: Explainable Metrics and Diverse Prompt Templates"
_ICLR.cc/2025/Workshop/BuildingTrust — BuildingTrust_

### Official Review · Reviewer_wm1G · 2025-02-17
**Accept**

**Rating:** 8
**Confidence:** 4

**Review:**

Many alignment experiments, such as in scalable oversight, depend on the use of LLM judges to simulate human judges. However there is a breadth of literature (which they cite) demonstrating that LLM judges have systematic biases such as position bias, length bias and self-bias that are different from human judges.

The paper attempts to give some solid theoretical grounding for formalizing and evaluating such biases, and provides a systematic evaluation framework for computing the accuracy and position and length bias of different LLM judges.

I think this paper gives a starting point to a very important field of study, and will likely be quite influential for future works involving LLM judges, such as in scalable oversight.

Some comments:

 (1) There should probably be some discussion on the natural or obvious pipelines that mitigate these biases, e.g. prompting the LLM with both orderings/a range of lengths and taking the average. You could also imagine a setup like AlphaGo: training an LLM on the output of such a pipeline to get it to “naturally” learn to be unbiased. Also with regards to general prompt biases, maybe dspy is something relevant?

(2) A quite related work is “consistency checks for language model forecasters”: https://openreview.net/forum?id=r5IXBlTCGc which discovers that LLM judges change their answers based on logical transformations of the questions (e.g. P → ¬P).

(3) Are accuracy, bias etc. defined only relative to human behaviour? i.e. an LLM is accurate if it gives the same answer as a human? This might make sense for the case of using LLM judges to simulate humans, but I think some discussion of the case where we are allowed to say the human is wrong, or the human would change their mind after encountering some new information, is valuable.

(4) Another area where the results of this paper may be valuable is: “Redesigning Information Markets in the Era of Language Models” https://openreview.net/forum?id=Zq9Dfj4nBo where you need language models to simulate their “human principals“ (humans who want the language model to make purchases on their behalf)

Overall, this is a clear accept. Perhaps the experiments are a bit small/weak, but this is forgivable for a workshop, and in any case the theoretical contribution is valuable enough.

---

### Official Review · Reviewer_9ATn · 2025-02-23
**Accuracy issues due to certain assumptions**

**Rating:** 5
**Confidence:** 2

**Review:**

**Strengths:**
1. The paper tackles the reliability of LLM-as-a-Judge methods, particularly for evaluating alignment techniques like RLHF and DPO.
2. The authors propose interpretable metrics for position bias and length bias, unifying them under an accuracy-based framework. The explicit modeling of self-inconsistency as flipping noise is novel.
3. Experiments highlight the significant influence of prompt templates on LLM judge performance, a relatively underexplored area.

**Weaknesses:**
1. The paper models LLM self-inconsistency as flipping noise with probability q, assuming q is independent of the true label X. This assumption may not hold, as LLMs might be more likely to flip decisions when uncertain, which often occurs when X = 0 (incorrect decision). The de-noising formula, e.g.,
\begin{equation}
p[X = 1 \mid (y_c, y_r)] = \frac{p[Z = 1 \mid (y_c, y_r)] - q_{cr}}{1 - 2 \cdot q_{cr}}
\end{equation}
relies on this assumption. If $q \mid X = 0 \neq q \mid X = 1$, the de-noised probabilities may be inaccurate.

2. The definition of position bias PB is tied to accuracy, as $\text{Acc}\_{\text{both}} = 1$ requires consistency across both response orders, implying zero position bias for that sample. The observed negative correlation between $\text{Acc}\_{\text{both}}$ and $|\text{PB}|$ may be tautological, as judges with high $\text{Acc}\_{\text{both}}$ must have low PB by definition.

3. Similarly, the claim that PB is "intrinsically length bias-mitigated" is questionable. If the LLM has a length bias, swapping positions could interact with length preferences, especially if the LLM favors longer responses in the first position.

---

### Official Review · Reviewer_gu3c · 2025-03-01
**Great insights for real-world LLM-as-a-judge uses but clarifications needed on some important points**

**Rating:** 6
**Confidence:** 4

**Review:**

Strengths:

- The authors introduced a set of reliability metrics of LLM-as-a-judge models with improved theoretical interpretability.
- The work can be useful in the real world as it demonstrated how users could mitigate some of the existing issues in LLM judges’ output in a principled way.

Weaknesses:

- Line 176 should be “$y_r$ more frequently in ($y_r$, $y_c$)”
- It is unclear what “intrinsically length bias-mitigated” means in Finding 1 at line 232. Same for “entangled with position bias” in Finding 2 at line 233. It’s also unclear how employing $A_\text{both}$ for accuracy can help mitigate the influence of positional bias. The authors are encouraged to add in more context or examples in the main text to provide better intuition to help readers understand.
- The authors mentioned at line 385 that “$A_\text{random}$ is less effective metric for assessing LLM judge performance compared to $A_\text{both}$”, and also simply switched to using only $A_\text{both}$ for evaluation in all the following sections. It leaves the readers wondering what the value of introducing $A_\text{random}$ in the paper really is? The authors need to clarify.
- Visibility of Figure 2, Figure 3, and Figure 4 needs to be improved.
- Do the results generalize beyond GPT models? The results can be quite limiting as the authors only looked at models from the same family.
- It’d be nice if the authors can provide some empirical evidence and/or case studies to showcase the dynamic between the position bias, the length bias, and $A_\text{both}$ as described in Finding 1 and Finding 2. I found that missing in the main text.
- One of the major contributions that the authors claimed is their open-source evaluation framework, but the link is not provided in the paper, which makes it hard to evaluate the quality of the actual framework.

---

### Decision · Program_Chairs · 2025-03-04

Accept